# Training and Competition Readiness in Triathlon

**DOI:** 10.3390/sports7050101

**Published:** 2019-04-29

**Authors:** Naroa Etxebarria, Iñigo Mujika, David Bruce Pyne

**Affiliations:** 1Research Institute for Sport & Exercise, University of Canberra, Bruce ACT 2601, Australia; david.pyne@canberra.edu.au; 2Department of Physiology, Faculty of Medicine and Nursing, University of the Basque Country, Leioa 48940, Basque Country, Spain; inigo.mujika@inigomujika.com; 3Exercise Science Laboratory, School of Kinesiology, Faculty of Medicine, Universidad Finis Terrae, Santiago 7501015, Chile

**Keywords:** health, periodization, intensity, concurrent training, fatigue, quantification, monitoring, nutrition

## Abstract

Triathlon is characterized by the multidisciplinary nature of the sport where swimming, cycling, and running are completed sequentially in different events, such as the sprint, Olympic, long-distance, and Ironman formats. The large number of training sessions and overall volume undertaken by triathletes to improve fitness and performance can also increase the risk of injury, illness, or excessive fatigue. Short- and medium-term individualized training plans, periodization strategies, and work/rest balance are necessary to minimize interruptions to training due to injury, illness, or maladaptation. Even in the absence of health and wellbeing concerns, it is unclear whether cellular signals triggered by multiple training stimuli that drive training adaptations each day interfere with each other. Distribution of training intensity within and between different sessions is an important aspect of training. Both internal (perceived stress) and external loads (objective metrics) should be considered when monitoring training load. Incorporating strength training to complement the large body of endurance work in triathlon can help avoid overuse injuries. We explore emerging trends and strategies from the latest literature and evidence-based knowledge for improving training readiness and performance during competition in triathlon.

## 1. Introduction

Triathlon is characterized by the multidisciplinary nature of the sport where swimming, cycling, and running are completed sequentially within the same event. The sport has a wide array of event formats, ranging from the mixed relay race (about 20 min), to the sprint distance race, lasting about 1 h, and the long-distance triathlon (Ironman), raced over an 8–9 h period at the elite level. In addition to the high training volumes typically undertaken for endurance sports, training for three different sporting disciplines simultaneously requires thoughtful planning of a large number of training sessions every week [1,2]. Large volumes of training can increase the incidence of illness and injuries, however, recent advances in knowledge in this area can minimize this risk while maximizing performance. This review examines the physiological (and biochemical) challenges of simultaneous multidisciplinary training and health risks associated with triathlon, individualized periodization and training strategies, and emerging trends in triathlon preparation.

The various formats and distances of triathlon racing all have their own discrete demands for different competition schemes. For example, in the main Olympic distance triathlon competition, a high level of sustained performance throughout the season is required, as the World Triathlon Series (eight events in 2019) reward the most consistent high-performing athlete with a World Champion title. In contrast, the long-distance events, particularly the Ironman, demand a single stellar performance on the day, given the very small number of races a triathlete usually undertakes in a year and the grueling physical demands of the lengthy race. Finally, there is the newest addition to the Tokyo 2020 Olympics program, the mixed relay race where two male and two female athletes complete a super-sprint triathlon—300 m swim, 6.6 km bike, and 1 km run—before tagging off to a teammate. A rather short and intense performance display for a so-called endurance athlete. The intricacy of triathlon goes beyond the multidisciplinary nature of the sport, and expands to athlete physical and mental health, training monitoring, nutritional strategies, and many other aspects. Careful integration of existing and emerging factors contributing to performance outcomes (Table 1) should promote adaptation to training, reduce the risk of injury and illness, and optimize training and competition readiness.

Triathletes sustain high training loads with various combinations of intensity and volume of training, represented by power output measured in watts, during cycling, for example (external load), and the associated perceptual measures and physiological responses (internal load), such as rating of perceived exertion (RPE) and heart rate (HR), blood lactate, and oxygen consumption. The uncoupling of internal and external loads is used to assess the fatigue status of an athlete [3]. For example, using the cycling external load mentioned above, the power output may be maintained for the same duration; however, depending on the fatigue state of the athlete, this may be achieved with a high or low heart rate or a high or low perception of effort [3].

The dissociation between an HR response (internal load) to a known low exercise intensity, such as 150 W in cycling (external load), whereby the HR response is elevated in response to the relatively low absolute intensity (external load), might reveal a marked state of fatigue in an athlete. To achieve optimal training progression leading to best race performance, various training-load monitoring tools have been developed to assist athletes and coaches in evaluating the readiness to perform, risk of illness and/or injury, and readiness to return to play from injury [4,5]. These athlete/training monitoring tools can highlight apparent disparities between internal and external loads and help the coach identify any looming problems before they materialize or are substantially aggravated.

In triathlon, as is common in most other sports, experience, anecdotal reports, and scientific facts are integrated to make informed decisions on training prescription. However, translation of research outcomes into individual training plans can be challenging as each athlete is different and can respond to training stimuli in different ways [6]. Work-to-rest ratios, injury and illness episodes, and magnitude of adaptations to training stimuli will all influence the coach’s decisions on individualized preparation for training and competition. Often, the best source for key information about optimizing training for athletes will come from the feedback provided by the athletes themselves [7]. Systematic athlete monitoring, anecdotal experience, and evidence-based knowledge will inform the coach to craft an integrated training plan individualized for each athlete.

### 1.1. Health First, Performance Follows

The primary aim of training is to prepare the triathlete for high-level competition. The journey to achieving this goal, however, will be different for most athletes. Individual requirements of frequency, volume, and intensity of training are different for each athlete, and an imbalance between training-induced fatigue and recovery can manifest in various ways. Some athletes suffer from excessive fatigue or overtraining, while others might succumb to injury or illness. During intensive training periods, carefully constructed individualized training plans should promote improvements in fitness capacities and performance, while avoiding setbacks. These setbacks are often caused by health-related issues (injuries) that follow sudden or abrupt increases or reductions in training loads [5]. Consistency in training is an important factor in optimizing the preparation process for competition, and an increased number of modified training weeks (due to illness or injury) can substantially reduce the chances of sporting success [8]. Adopting an integrated approach based on effective communication with a close relationship between the clinician (case manager) and the coach is a key for success [9].

Athlete training and athlete monitoring programs work in combination and typically incorporate training loads, health and well-being, physiological, dietary, and recovery strategies. A healthy immune system and a robust anatomical structure to avoid illness and injury are the foundations that support athletic training and competitive performance [8,10]. Most high-performing athletes will experience one or more significant health issues (or a sequence of them) that slow their progress in training at some point during a competitive season. The incidence of an injury per 1000 h of training has been reported as 0.7–1.4 during training and 9–19 during competition, most of which (50%) seems to derive from running, 43% from cycling, but only 7% with swimming [11]. Problems can take many forms from an acute injury, a more chronic condition that reaches breakpoint, or a temporary illness caused by sub-optimal nutritional intake, a long-haul travel-related episode, or the usual common cold. The long-distance triathlon requires a high intake of nutrients, especially carbohydrates, that can cause issues in the gastrointestinal tract [12,13]. Educating athletes and support staff for best practice in management of illness and preventative measures [14] is a major part of effective athlete health management (Table 1).

### 1.2. Multidisciplinary Training—Interfering or Additive?

A challenge for many endurance sports including triathlon is understanding how the cellular level signaling responds to multiple modes of training. For example, skeletal muscle from endurance- and strength-trained individuals have diverse adaptive states, and simultaneous training for both endurance and strength results in a compromised adaptation, compared with training for either exercise modality alone [15], a phenomenon called the “interference effect” [16,17]. It is unclear how much interference occurs when simultaneously training for the three disciplines of swimming, cycling, and running. On the other hand, when multiple training stimuli are aligned in terms of timing, recovery, and balance between intensity and volume, the additive effects can yield central and certain peripheral physiological adaptations. This occurs when adaptations from different exercise modes are transferred a response, referred to as cross-training [18,19]. Despite limited evidence, a triathlete’s running ability can improve from cycling-induced aerobic central adaptations and vice versa [20]. Maximizing the return from each training session by amplifying the biochemical pathways during training and recovery is a goal in any sport. This is especially the case in triathlon, where athletes deal with multiple disciplines that necessitate high to very high training loads. More research is required to fully understand the conjoined/simultaneous metabolic processes triggered by frequent training stimuli of varied duration, intensity, and exercising modes.

As sporting performances continue to improve, new strategies to maximize performance emerge, giving triathletes an edge in training and competition. In search of maximizing the training stimuli and consequent desired adaptations, the triathlete runs the risk of maladaptation. To avoid initiating metabolic pathways that might be detrimental to training progress, some basic understanding of metabolic signaling events is needed. Intra- and inter-individual variability in sports performance is largely due to metabolic flexibility and adaptation plasticity that underpin individual responses to training. Metabolic flexibility relies on the configuration of metabolic pathways that manage nutrient sensing, uptake, transport, storage, and utilization. This metabolic organization is mediated by synthesis, degradation, or activity regulation of key proteins or enzymes [21]. Metabolic flexibility underpins adaptation plasticity, accounting for substantial differences in the degree of adaptation or performance ability between individuals in response to the same training program. Adaptation plasticity is specific to the mode of exercise, timing, and individual responsiveness to different types of contractile activity [6]. However, peak induction for both metabolic and myogenic (muscle tissue) genes responsible for adaptation, generally occurs 4–8 h after an exercise bout. The mRNA (biologic messenger that translates exercise stimuli into anatomical, biochemical, and physiological adaptations) returns to pre-exercise levels within 24 h [22]. Triathletes undertake multiple training sessions a day, yielding a continuous overlay of molecular pathways in each 24 h window. Endurance training adaptations are dependent on the mode of exercise, the volume, intensity, and frequency of the contractile stimuli [23]. However, biological evidence to inform real world questions regarding volume, intensity, and timing of training stimuli for athletes is scarce. More understanding of these intracellular signaling cascades is needed to inform timing and sequence of training sessions for triathletes.

## 2. Training Periodization

The most important goal for coaches and triathletes is to maximize the competitiveness of the athletes, and design a well-controlled training program to ensure that peak performances are aligned with major triathlon competitions. Traditional training periodization, with its usual division of the training season into hierarchical preparatory, competitive, and transition periods, and structural components called macrocycles, mesocycles, and microcycles [24], provides coaches and athletes with basic guidelines for structuring and planning their training. In triathlon, top performances are often associated with periods of intensive training, followed by a taper, which involves a marked reduction in the training load for a few days before a major competition [25]. A taper intends to minimize a triathlete’s habitual stressors, allowing physiological systems to undergo supercompensation [26]. An overload training period immediately preceding a taper may elicit larger subsequent performance gains in highly trained triathletes, but not in the presence of excessive fatigue, which increases the risk of training maladaptation and infection [27]. 

Although traditional periodization may be a perfectly valid strategy for long-distance triathletes targeting two or three major races in a season, a major limitation of this approach is its inability to elicit multiple peaks for repeated racing over the competitive season [28]. Elite triathletes competing in Olympic distance events have fewer opportunities to taper because repeated consistent top-level race performance is a key feature of the sport’s competitive structure. Peaking strategies for multiple races will depend on the triathlete’s level of fatigue after a race, or series of races, and the time frame between triathlons [25]. Block periodization, characterized by the sequencing of highly specialized accumulation, transmutation, and realization mesocycle blocks, could be a suitable alternative to traditional periodization for attaining multiple fitness and performance peaks throughout a competitive season [28]. The biological underpinnings of block-periodized endurance training have been reviewed recently [29].

Whatever the periodization approach, training prescription should be aligned with contemporary elite practice and evidence-based conceptual models, together with previous experiences, observations, and data, allowing contextualized decisions and effective management of the training process [30]. In this respect, multiple periodized approaches can be used at various points of an athlete’s career or even within the same training season [31]. A flexible periodization strategy may also allow an Olympic distance triathlete to maintain high fitness throughout the season, which is often necessary to ensure high world and/or Olympic rankings. In this context, a world-class female triathlete was able to maintain a relatively high competitive level throughout an entire Olympic season (seventh place in the Triathlon World Ranking for 2012), and multiple fitness and performance peaks were achieved by means of planned training tapers in the lead-up to key international events [2].

A recent development in the topic of periodization is the concept of integrated periodization, which coordinates multiple training components best suited for a given training phase in an athlete’s program. This concept could well represent a step towards best practice in triathlon training. The available evidence underpinning integrated periodization was recently reviewed, focusing on exercise training, recovery, nutrition, psychological skills, and skill acquisition as key factors by which athletic preparation can be optimized [32].

### 2.1. Training Intensity Distribution

The majority of competitive endurance events are performed at intensities close to an athlete’s individual lactate threshold. However, observational studies on the training intensity distribution in various endurance sports, including swimming [33], cycling [34], running [35,36], and triathlon [2,31,37], show a strong focus on training at low-to-moderate intensities below the lactate threshold, with most of the remaining training time targeting high-intensity training at near-maximal and supramaximal intensities. This format of polarized training intensity distribution [38] is considered best practice to maximize adaptation at acceptable levels of physiological stress [39,40]. Well-trained endurance athletes show improvements in key variables related to endurance performance by manipulating training towards a polarized intensity distribution [41,42,43,44]. For example, a world-class female Olympic distance triathlete performed 74%, 88%, and 85% of her swim, bike, and run training, respectively, at intensities below her individual lactate threshold over an entire season [2]. Even higher percentages (82%, 91%, 88%) were reported in a world-champion long-distance paratriathlete [31]. In addition, faster Ironman performances are associated with longer training times at low-to-moderate intensities [37].

These somewhat paradoxical polarized training models may be explained by the greater effectiveness of both light and very intense exercise on aerobic phenotypic adaptations, linked to activation of intracellular signaling cascades. Upstream modulators of peroxisome proliferator-activated receptor-c coactivator (PGC) 1α expression influence mitochondrial biogenesis, oxidative phosphorylation, and other features of oxidative muscle fibers in skeletal muscle [45]. It has also been speculated that modern humans are physiologically better adapted to training modes similar to the exercise patterns that their hominid ancestors evolved on, which were mainly characterized by the prevalence of daily bouts of prolonged, low-intensity, aerobic-based activities, interspersed with periodic, short-duration, high-intensity bursts of activity [46].

### 2.2. Strength Training for Triathlon Performance

A well-planned and periodized strength training program should complement triathlon training throughout a season, allowing proper long-term athlete development, limiting the risk of injury, and eventually maximizing competition performance. Indeed, most elite triathletes nowadays combine their long, mostly aerobic swim, bike, and run training sessions with some form of strength training (i.e., concurrent training). Given the wide range in duration of the various triathlon events (e.g., ~approximately 20 min for an elite athlete racing in the mixed relay vs. approximately 8 to 9 h for elite male and female Ironman athletes, respectively), and the selective contribution to performance of upper- and lower-body muscle groups, both aerobic endurance and muscle strength are important to enhance competitive triathlon performance.

A recent meta-analysis highlighted the benefits and supported the implementation of strength training to complement the sport-specific aerobic training in middle- and long-distance events, irrespective of sport and level of athlete [47]. During the swim leg of a triathlon race, for instance, upper-body muscular strength and power should translate into increased ability to generate propulsive force in the water, improved stroke length and/or stroke rate, and increased free swimming speed [48]. Therefore, both dry-land and in-water strength training can be beneficial to performance during a triathlon swim [49]. Lower-body strength and power training can improve cycling ability and time trial performance [50,51] by reducing oxygen uptake, HR, blood lactate concentration, and RPE during prolonged cycling [52], and by eliciting an earlier peak torque during the pedal stroke [51]. Lower-body explosive strength training [53] and plyometric training [54] can also enhance running economy and performance. Well-trained triathletes performing a heavy weight training program in combination with their usual endurance training improved their maximal aerobic velocity, running economy, and hopping power [55], and delayed the onset of fatigue during prolonged submaximal cycling [56]. The beneficial effects of strength training on cycling and running economy and performance are confirmed by recent meta-analyses [47,57,58,59].

Greater effects on performance are yielded as a result of periodized heavy strength programs designed for maximal force development (e.g., 2–3 sets of 4 to 10 repetition maximum), involving sport-specific muscle groups and movements, focusing on performing the concentric phase of the lifts with maximal intended velocity, via two sessions per week for 12–24 weeks [47,60]. The improved endurance performance may relate to delayed activation of less efficient type II fibers, improved neuromuscular efficiency, conversion of fast-twitch type IIX fibers into more fatigue-resistant type IIA fibers, or improved musculo-tendinous stiffness [60], with no detrimental effects on maximal oxygen uptake and other markers of aerobic endurance [47,55]. In addition, strength training is considered the most effective exercise intervention to prevent overuse injuries in sport [61].

### 2.3. Quantification of Training and Competition Loads

Load is defined as the sport- and non-sport-related burden (single or multiple physiological, psychological, or mechanical stressors) as stimuli that are applied to a human biological system (including subcellular elements, a single cell, tissues, one or multiple organ systems, or the individual) [62]. Load can be applied to an athlete over varying time periods (seconds, minutes, and hours to days, weeks, months, and years) and magnitudes (i.e., duration, frequency, and intensity) [62]. Accurate and reliable quantification of the training load undertaken by an athlete is necessary to analyze and establish causal relationships between the training performed and the resultant physiological and performance adaptations [63,64,65]. Whatever the quantification methods used, they can be defined as quantifying either external or internal training load [3,66]. The external training load, which is measured independently of the internal workload [66], is an objective measure of the work that an athlete completes during either training or competition (e.g., hours of training, distance run, power output produced). However, other external factors, such as life events, daily hassles, or travel, may be equally meaningful [62].

In contrast the internal workload assesses the biological and psychological stress imposed by the training session, and is defined by disturbance(s) in homeostasis of the physiological and metabolic processes during the exercise training session [66]. Specific examples include measures such as HR (physiological/objective), RPE, or inventories for psychosocial stressors (psychological/subjective) [62] A recent study on the relationships between various training load measures in professional cycling showed that load measures based on RPE, HR, and power output are all reliable for quantifying training load in training and racing, and concluded that any method of training load quantification, which is consistently applied and discussed between coach and athlete may be equivalent in net value [67]. Several reports are available that summarize the most relevant workload quantification methods in long-duration cyclic sports [62,63,68]. Practical examples of their applications have been provided to adjust the training programs of elite athletes in accordance to their individualized stress/recovery balance [65]. It is noteworthy that a triathlon-specific load quantification method is available, which combines objective and subjective load coefficients and discipline-specific (i.e., swim, bike, run) weighting factors, but this method requires further scientific validation [69].

### 2.4. Monitoring Fatigue and Adaptation

Elite athletic preparation requires a fine balance between pushing the training and adaptation boundaries for performance, and avoiding negative outcomes, such as underperformance, injury, illness, or poor well-being [3,70]. Inappropriate loading may lead to excessive accumulated fatigue and maladaptive processes and increase injury risk by impairing factors, such as decision-making ability, coordination, and neuromuscular control. Excessive fatigue can contribute to increased risk of acute and overuse injuries [62]. The measurement and monitoring of fatigue and recovery in training and competition is a complex task requiring expertise in physiology, psychology, and sport science [71].

Elite triathletes’ training loads can be extremely demanding, e.g., an average of 16 weekly sessions, with only 21 days of full rest over a 50-week Olympic season [2]. No single marker of an athlete’s response to load consistently predicts maladaptation or injury [63], and a combination of external and internal load markers, subjective and objective measures is generally considered best practice. For instance, multiple biomarkers reflecting an athlete’s positive adaptation or maladaptation to periods of intensive training demonstrate inconsistent findings, due in part to large inter-individual variability [72]. A systematic review of objective and subjective measures of athlete well-being to guide training and detect any progression toward negative health outcomes, and associated poor sports performance, indicated that athletes should report their subjective well-being on a regular basis (ideally daily), alongside other monitoring practices [68]. A multivariate approach, including physiological, biomechanical, cognitive, and perceptive monitoring, has been recommended to prevent maladaptation in highly trained triathletes [73]. Similarly, a recent study on the monitoring of professional road cyclists and elite swimmers during training camps provided further support for a multi-faceted approach to monitoring fatigue, recovery, and adaptation [70]. Furthermore, focusing on individual rather than group responses [72] and/or comparing individual to group day-to-day change in monitored variables may prove effective in flagging athletes potentially at risk of maladaptation [70].

## 3. Emerging Trends in Triathlete Preparation

Similar to other sports in the professional era, the competition structures, approaches to training, and support services in triathlon continue to evolve. While some national programs and professionally supported triathletes benefit from a well-resourced training and competition support program, most clubs and individual triathletes must take responsibility for managing their own preparations. While many evidence-based preparation strategies and practices are well established, new approaches continue to emerge as nations, programs, and individual coaches and athletes continually seek a competitive edge. Here, we highlight emerging trends in triathlon (and high-performance sports) that directly or indirectly influence training and competitive performance (see Table 2).

### 3.1. Psychological Factors

The traditional approach has in part focused on performance psychology to assist the athlete in their sporting pursuits. The psychologist has traditionally been more clinically oriented with only the occasional foray into the competition and training domains. Athletes often only sought assistance from a sports psychologist after some issue presented itself that caused problems at a personal or group level. In more recent times, psychology has broadened substantially to address a wide range of settings and issues. Mental toughness is often seen as an important factor and it appears that triathletes get stronger in this regard as they mature and obtain more race experience [74]. Another promising area is how athletes cope with mental fatigue arising from high level training and competition, and/or in combination with other lifestyle stresses [75]. Future work in mental fatigue will identify strategies for improving the ability of athletes to meet both the acute and chronic mental demands of high-volume training. Until recently, mental health issues were largely dealt with individually (in a private setting) and rarely in a team or public domain. Psychological skills training is increasingly undertaken proactively by athletes, while mental health issues are being dealt with in more confidence [76] and, in some high profile cases, chronicled in the public domain. It should be emphasized that sports and physical activity can be a positive factor in health-related quality of life in college-age individuals [77], so it is a matter of balancing the positive and negative factors of high level involvement in triathlon.

### 3.2. Training

Although the demands of training and competition are well understood, there are still coaches and athletes wedded to the more-is-better training philosophy. Progressions in training load should be periodized rather than strictly linear in nature, whatever the race distance. A range of external training load factors and baseline characteristics have been associated with an increased rate of injury and/or pain in endurance sports [78]. In contrast, more work is needed on effective markers of internal training loads. Somewhat contrary to common perception, the relationships between training volume and injury are more complex than a simple linear relationship between risk factors and occurrence of common injuries [79]. Work is now progressing on more uniform definitions and terms, better measures of internal and external training loads, and more sophisticated data analytics to improve the understanding of relationships between training and injury/illness risk. This information is needed to update current knowledge and prepare practical guidelines for the triathlon community.

### 3.3. Nutrition

Traditionally the focus of athlete nutrition has been on absolute and relative macro- and micro-nutrient intake. Triathlon has focused on carbohydrate intake, given its importance in fueling endurance training and competition formats, such as the Olympic distance and Ironman events. More recent work has highlighted the importance of timing of nutrient intake in relation to training and competition. New strategies are being promoted, such as “sleep low”, which involves a sequential periodization of carbohydrate (CHO) availability and low glycogen recovery after “train high” glycogen-depleting interval training, followed by an overnight-fast and light intensity training (“train low”) the following morning. [80]. In contrast, chronic ketogenic low-CHO high-fat diets might impair iron metabolism, aspects of immune function [81], performance, and well-being [82]. Further work is in progress to identify how diets can be individualized according to event demands, athlete background, training demands, and whether changes in body composition are required to improve performance.

During the 1990s and 2000s, the so-called female athlete triad was the predominant exercise model accounting for health issues in female athletes. The condition was characterized by athletes presenting with low energy availability, menstrual dysfunction, and low bone mineral density. In recent years, the relative energy deficiency (REDs) term has emerged, recognizing that low energy availability affects both female and male athletes, and a broader range of health and performance parameters, not just bone health and menstrual dysfunction [83]. Challenges around accurate clinical or laboratory measurement of energy availability, the perennial issue of body mass and composition management, and effective education strategies are currently being addressed. Further developments in this area will assist the goal of reducing the prevalence and incidence of injury and illness.

Heat stress and fluid replacement strategies are issues often faced during triathlon competition under hot and/or humid environmental conditions, particularly in the long-distance and Ironman formats. The benefits of carbohydrate content and fluid volume in sports drinks have been studied extensively and triathletes should pay particular attention to these matters [84]. However, greater reductions in body mass and higher post-competition core temperatures have been recorded in faster triathletes, indicating these competitors can push themselves harder and/or tolerate the effects of sweat loss and heat more effectively [85]. Investigators are continuing to develop innovative methods, such as ice slurry ingestion [86], new sports drink formulations, and manipulating drink content and timing before, during, and after training and competitive events, especially in the important hours after heavy exertion.

### 3.4. Clinical/Medical

Medical management is evolving from a healthcare and provider (medical doctor)-centered system, with a treatment focus and paper records that made consistency and retrieval of individual medical records difficult, to an athlete-centered system. New systems are evolving with an injury and illness prevention focus and personalized medicine, using the full suite of digital and technological solutions and systems. Improvements are also likely to come in the areas of improved biomedical testing (in immunological, oxidative stress-related fatigue and cardiovascular markers), improved clinician diagnoses, and field-based studies of race-related injuries and illnesses [87]. Personalized predictive medicine with a focus on genetics has arrived in clinical medicine, but will require additional metadata and biological validation to identify a comprehensive set of genes useful in sports [88]. Perception of injury and training risk factors among health professionals center primarily on training load and demographic characteristics. In one study, three common factors accounted for over 50% of the variance in injury risk in triathletes: The underlying training, health and medical monitoring, and preparation of the triathlete for competition [89]. This information points to the critical factors of training, monitoring, and competition preparations, all of which inform the upskilling of practitioners and training of the next generation of sports professionals.

### 3.5. Lifestyle Behaviours

Treatment and management priorities typically arise after an athlete has succumbed to fatigue, injury, or illness. In the future, however, increased attention will be given to prevention strategies, with athletes taking more self-responsibility for training, recovery, sleep, nutrition, and other factors. Sleep management or sleep hygiene is now a major focus of athlete preparation and research is driving new innovations and interventions to improve this important factor. Strategies for managing travel stress and jet lag should be implemented by triathletes embarking on long haul flights. Approaches for heat acclimation training and altitude training also continue to evolve. Hard-earned experience of athletes and coaches, and the results of research investigations, will generate more nuanced scheduling of heat and altitude training. Key themes for increasing the benefit of altitude and heat interventions include more effective preparatory training in the weeks before, implementation of more useful internal and external load measures, and nutritional strategies that maximize the adaptations needed to enhance performance at sea-level and altitude, and in temperate and hot environments [90].

### 3.6. Coordination

International federations, such as the International Triathlon Union (ITU), and national federations have traditionally managed competition and travel schedules; organized training, programs and tours; provided medical and scientific support both in domestic and international settings; and conducted coach–athlete education programs. This work requires substantial resourcing and policy development. The increasing professionalism, commercial funding, and sponsoring of programs, teams, and individuals has markedly changed the management and control of athlete programs and competition preparations. While this work in the organizational and management areas will continue, there is increasing need for clearer and more effective translation of expert knowledge (including coaches, triathletes, and support staff) and implementation of technology, research outcomes, and other improvements and innovations into national programs, clubs, competitions, and everyday training. Translation and innovation will require cooperation and communication between governing bodies, athlete and coaching groups, research entities, and across national borders.

### 3.7. Research

Despite the popularity of triathlon, the sport has received much less attention from industry and academic researchers than cycling, running, and swimming. Coaches and scientists in triathlon have to translate the outcomes from other sports to improve the management and performance of triathletes [91,92]. In the future, more triathlon-specific research will be conducted to promote best practice in the sport in junior, senior, and elite competitors. Triathletes and coaches will be more involved in research, rather than projects being driven largely by scientists and/or academic researchers. Like other sports, the focus of research is evolving from specific disciplines (for example, psychology, performance analysis, physiology, nutrition, medical and allied health) driven by scientists to multi-disciplinary research, fully integrating the coach and athlete. There will be more focus on technological innovation and sophisticated data analytics of training management and race performances.

## Figures and Tables

**Table 1 sports-07-00101-t001:** General guidelines for illness prevention in athletes; adapted from Schwellnus et al. [14].

Behavioral, Lifestyle, and Medical Strategies	
Athletes are Advised to:	Minimize contact with infected people, young children, and animals; Avoid crowds and minimize contact with people outside the team/support staff; Keep at a distance to people who are coughing, sneezing, or have a “runny nose”; Wash hands regularly and effectively with soap and water, especially before meals; Carry insect repellent, antimicrobial foam/cream, or alcohol-based hand washing gel; Not share drinking bottles, cups, cutlery, towels, etc., with other people; Choose beverages from sealed bottles, and avoid raw vegetables and undercooked meat; Wear open footwear when using public showers and swimming pools; Adopt strategies to facilitate good quality sleep at night and nap during the day.
Support Staff are Advised to:	Develop, implement, and monitor illness prevention guidelines for athletes and support staff; screening for airway inflammation disturbances (e.g., asthma, allergy); Identify high-risk athletes to take precautions during training/competition; Arrange for single-room accommodation during competition; Update athletes’ vaccines needed at home and for international travel.
**Training and Competition Load Management**	
Poor load management with ensuing maladaptation can be a risk factor for acute illness and overtraining. Changes in training load should be individualized in small increments <10%. General recommendations are:	Detailed training/competition plan, including post-event recovery strategies; Training load monitoring, using measurements of external and internal load; Adequate nutrition, hydration, sleep, relaxation strategies, and emotional support.
**Psychological Load Management**	
Psychological load (stressors) such as negative life event stress and daily hassles can increase the risk of illness in athletes. Clinical practical recommendations center on reducing state-level stressors and educating athletes, coaches, and support staff in proactive stress management:	Develop resilience strategies that help athletes manage negative life events, thoughts, emotions, and physiological states; Education for stress management techniques, confidence building, and goal setting; Reduce training/competition loads after negative life events to mitigate risk of illness; Implement periodical stress assessments.
**Measuring and Monitoring for Early Signs and Symptoms of Illness Over-Reaching and Overtraining**	
An athlete’s innate tendency is to continue to train and compete despite physical complaints or functional limitations. It is recommended that:	Ongoing illness (and injury) surveillance systems should be implemented; Athletes be monitored for subclinical signs of illness, such as non-specific symptoms; Athletes be monitored for early symptoms and signs of over-reaching or overtraining.

**Table 2 sports-07-00101-t002:** Emerging new concepts in endurance training and triathlon to minimise fatigue, illness, and injury.

Factor	Traditional View	Emerging Trends
Psychological	Acute events focus Discipline seen as clinical Minimal athlete education No mental health priority	Integrated model Psychological skills training Mental health
Training	More is better philosophy Rudimentary training monitoring	Event formats dictating preparation Load is not linear Sophisticated training monitoring Integrated periodization approach
Nutrition	Macro- and micronutrient intake are important Female Athlete Triad (low energy availability, menstrual dysfunction, and low bone mineral density)	Timing on intake in relation to training and competition (i.e., periodized sports nutrition) Relative energy deficiency (REDs) New drink formulations and event-specific ingestion
Clinical/medical	Healthcare provider-centered system Treatment focus Paper records	Athlete-centered system Prevention focus Personalized medicine Digital focus
Lifestyle factors, including hygiene, travel, sleep	Competition focus Training load focus Treatment/management focus Team responsibility	Prevention or prophylactic focus Self-responsibility Travel management emphasized Sleep management focus More nuanced scheduling
Coordination	Policy Position statements Guidelines	Translation into practice Implementation
Research	Limited triathlon studies Discipline-specific Scientist-driven	Triathlon-specific Multi-disciplinary teams Coach athlete involvement more clearly defined Increasing technological involvement More sophisticated data analyses

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
