# Peer review of "Training and Competition Readiness in Triathlon"

_sports, 2019, doi:10.3390/sports7050101_

Round 1
Reviewer 1 Report
With interest I have read your review ”Training and competition readiness in triathlon”.
This work aims to explore emerging trends and strategies from the latest literature and evidence-based knowledge for improving training readiness and performance during competition in triathlon.
The work presents a quite interesting topic for the target audience of the “Sports”, particularly for the Special Issue “Maximising Triathlon Health and Performance: the State of the Art”.
Generally the review is well conducted. The writing of the article is of very good quality. However, there are some challenges within the organization and methodical approach.
Below, I would like to comment on the authors study in respect to their performed work.
I have some major reservations with the paper which reduce enthusiasm.
1. In the whole manuscript, there is no adequate differentiation between the different distances. However, nearly in all sections that are given (i.e. training, monitoring, medical….) there are essential differences that should be provided.
2. The manuscript is too long and some sections appear superficial and do not add new or relevant findings to the existing knowledge.
Minor concerns:
A methods section is missing. Although I fully agree with a narrative review in this field, I recommend adding a short methods section. The reader should be informed how the review was conducted.
Line 24: “triathlon”. Which kind of ‘triathlon’ do you mean and which kind of triathlon are you willing to address? All distances and disciplines?
Line 45: The reviewer agrees that MTR are new in an Olympic program to Tokyo 2020. However, MTR races are currently performed in elite races (i.e. WTS series) for some years. Please clarify
Line 61: Please provide a reference
Line 77-106: This section should be shortened in a significant manner.
Line 159-196: Please shorten. However, different physiological demands in the different disciplines should be given clearly in terms of intense, loads, metabolic power…
Line 285-297: Please shorten
Line 345-358: Please shorten
3.3 Nutrition: The topic of fluid and electrolytes should be provided
Author Response
Dear Editor,
The authors sincerely thank the reviewers for their assessment and valuable comments on
our manuscript titled ‘Training and competition readiness in triathlon’.
We have responded in a point-by-point format to each comment and where appropriate have revised the manuscript accordingly. Changes made in the revised version of the manuscript appear in blue text. The reviewers’ comments are shown in black, and our responses in blue.
Reviewer 1:
With interest I have read your review “Training and competition readiness in triathlon”.
This work aims to explore emerging trends and strategies from the latest literature and evidence-based knowledge for improving training readiness and performance during competition in triathlon.
The work presents a quite interesting topic for the target audience of the “Sports”, particularly for the Special Issue “Maximising Triathlon Health and Performance: the State of the Art”.
Generally the review is well conducted. The writing of the article is of very good quality. However, there are some challenges within the organization and methodical approach.
Below, I would like to comment on the authors study in respect to their performed work.
Thank you very much for your feedback.
I have some major reservations with the paper which reduce enthusiasm.
1. In the whole manuscript, there is no adequate differentiation between the different distances. However, nearly in all sections that are given (i.e. training, monitoring, medical....) there are essential differences that should be provided.
Thank you for your feedback. We have specifically mentioned the discipline of triathlon where a certain concept or recommendation might apply directly, where appropriate. For example the new paragraph within the Nutrition section which includes the following: ‘Heat stress and fluid replacement strategies are issues often faced during triathlon competition under hot and/or humid environmental conditions,
particularly in the long-distance and Ironman formats’. Other sections also follow a similar example.
In lines 160 – 164 which read as follows: ‘Although traditional periodization may be a perfectly valid strategy for long-distance triathletes targeting two or three major races in a season, a major limitation of this approach is its inability to elicit multiple peaks for repeated racing over the competitive season (28). Elite triathletes competing in Olympic distance events have fewer opportunities to taper because repeated consistent top-level race performance is a key feature of the sport’s competitive structure.’
In lines 182 – 183 which read as follows: ‘This type of flexible periodization strategy may also allow an Olympic distance triathlete to maintain high fitness throughout the season, which is often necessary to ensure high world and/or Olympic rankings.’
In lines 361 – 362 which read as follows: ‘Triathlon has focused on carbohydrate intake given its importance in fuelling endurance training and competition formats such as the Olympic distance and Ironman events’
2. The manuscript is too long and some sections appear superficial and do not add new or relevant findings to the existing knowledge.
Minor concerns:
A methods section is missing. Although I fully agree with a narrative review in this field, I recommend adding a short methods section. The reader should be informed how the review was conducted.
Thank you for your feedback. As a narrative review that incorporates new and emerging insights into the sport, rather than merely reviewing the existing literature on a specific topic, we have incorporated the newest findings in research (from publications in 2017-2019) that are applicable to triathlon. We consider that the narrative review gains value by the extensive coverage of many different research areas rather than completing a comprehensive review of a narrow research topic. The nature of the review limits opportunity for a methods section.
Line 24: “triathlon”. Which kind of ‘triathlon’ do you mean and which kind of triathlon are you willing to address? All distances and disciplines?
As per the explanation above, most of the concepts presented in the review apply (to a greater or lesser degree) too all disciplines of triathlon and where a specific idea or concept is more relevant to a certain discipline or race duration (triathlon event), the authors have made reference to it.
Line 45: The reviewer agrees that MTR are new in an Olympic program to Tokyo 2020. However, MTR races are currently performed in elite races (i.e. WTS series) for some years. Please clarify
The authors mention that this new discipline will be part of the official Olympic program in Tokyo for the first time, and do not argue the race has not been operational for a few years.
Line 61: Please provide a reference.
Thank you. The authors have now added a reference (Halson, 2014) to the statement.
Line 77-106: This section should be shortened in a significant manner.
Thank you. We have now shorten this section.
The following sections have been cut to make the paragraphs more succinct:
in managing athlete health, especially for athletes with existing or ongoing health issues, is important to achieve consistency of training for elite athletes. Having
, which are often the main health challenge for athletes during training and competition
A healthy body and mind are paramount to excel on the field, especially for triathletes who sustain training loads in three different sporting disciplines.
Line 159-196: Please shorten. However, different physiological demands in the different disciplines should be given clearly in terms of intense, loads, metabolic power...
Thank you. The text has been significantly shortened and revised as suggested. The deleted section is below: major competitions every two to four weeks poses the challenge of having to choose between allowing sufficient recovery from a previous training-competition cycle, and then rebuilding fitness, or maintaining intensive training to capitalize on adaptations acquired during the previous cycle. Both approaches can be valid, depending
Line 285-297: Please shorten
Thank you for this comment. We have shortened the paragraph as suggested. The deleted section is below: This approach is characterized by the sequencing of highly
specialized accumulation, transmutation, and realization mesocycle blocks. Block periodization is intended to prevent conflicting physiological responses between non- compatible training stimuli, provide a favorable interaction for training effects, and benefit from the favorable interaction of cumulative and residual training effects
The training of a world-champion long-distance paratriathlete, for instance, was organized as a traditional periodization macrocycle in the first half of each season, followed by a block-periodization approach for the remainder of the season
Line 345-358: Please shorten
This paragraph has now been shorted as suggested.
The deleted section is below: The biological adaptations induced by adequate training loads increase an athlete’s capacity to withstand further load, facilitate progressive adaptations and performance improvements, and enhance resilience to injuries
reduce muscular force development and contraction velocity, which in turn can increase the forces imposed on passive tissues, adversely alter kinetics, kinematics and neural feedback, reduce joint stability, and
3.3 Nutrition: The topic of fluid and electrolytes should be provided
Thank you for your pertinent suggestion on fluids and rehydration issues. We have now added the following paragraph and 3 new references to expand the ‘nutrition’ section as suggested by the reviewer.
Heat stress and fluid replacement strategies are issues often faced during triathlon competition under hot and/or humid environmental conditions, particularly in the long-distance and Ironman formats. The benefits of carbohydrate content and fluid volume in sports drinks have been studied extensively and triathletes should pay particular attention to these matters (84). However, greater reductions in body mass and higher post-competition core temperatures have been recorded in faster triathletes, indicating these competitors can push themselves harder and/or tolerate the effects of sweat loss and heat more effectively (85). Investigators are continuing to develop innovative methods such as ice slurry ingestion (86), new sports drink formulations, and manipulating drink content and timing before, during and after training and competitive events, especially in the important hours after heavy exertion.
We have also added one new dot point In Table 2. cross-reference to the Nutrition section as follows: New drink formulations and event-specific ingestion
Reviewer 2 Report
The authors present a review about general aspects how to prepare for competition and training.
The manuscript is valuable, because it reviews a high number of triathlon relevant manuscripts. The presented work is helpful for athletes, coaches and the medical staff.
Training between sprint and long distances is often different and should be discussed separate.
The abstract is sufficient.
Introduction:
Leads to the topic. Table 1 is not highly scientific, but helpful for daily life in professional sports. Adding table 1 as supplementary material would be enough.
I would recommend explaining at the end of the introduction, what is the aim of the review.
1.1. Health First, Performance Follows:
Is sufficient and makes sense.
1.2. Multidisciplinary training – Interfering or Additive?
Is interesting to read, authors state correctly, that not much is known about the molecular pathways.
2. Training periodization:
Interesting to read. Good overview about the specific literature.
3. Emerging Trends in Triathlete Preparation
Table 2 is particularly helpful.
Author Response
Reviewer 2:
The authors present a review about general aspects how to prepare for competition and training.
The manuscript is valuable, because it reviews a high number of triathlon relevant manuscripts. The presented work is helpful for athletes, coaches and the medical staff.
Training between sprint and long distances is often different and should be discussed separate.
The abstract is sufficient. Introduction:
Leads to the topic. Table 1 is not highly scientific, but helpful for daily life in professional sports. Adding table 1 as supplementary material would be enough.
Thank you – the authors agree but as journal articles don’t usually have appendixes, we decided it could be part of the manuscript.
I would recommend explaining at the end of the introduction, what is the aim of the review.
Thank you for the suggestion. At the end of the first paragraph with the introduction we state what the aim of the review is: ‘This review examines the physiological (and biochemical) challenges of simultaneous multidisciplinary training and health risks associated with triathlon, individualized periodization and training strategies, and emerging trends in triathlon preparation.’ The introduction introduces the sport and the generic aspects that are key to the review, as the first paragraph of the introduction introduces the review paper with a statement of what to expect from the review.
1.1. Health First, Performance Follows: Is sufficient and makes sense.
Thank you.
1.2. Multidisciplinary training – Interfering or Additive?
Is interesting to read, authors state correctly, that not much is known about the molecular pathways.
Thank you.
2. Training periodization:
Interesting to read. Good overview about the specific literature.
Thank you.
3. Emerging Trends in Triathlete Preparation Table 2 is particularly helpful.
Thank you.
Reviewer 3 Report
The proposed review summarizes major scientific and practical aspects related to preparation and competitive performance in triathlon. Taking into account deficiency of available information actuality and importance of this paper cannot be underestimated. The authors highlight the crucial issues such as interference of training workloads, intensity distribution, training periodization and fitness training. Emerging new concepts related to training, health status and injuries prevention were presented and considered. The proposed review has distinct value both for sport science and practice.
Author Response
The proposed review summarizes major scientific and practical aspects related to preparation and competitive performance in triathlon. Taking into account deficiency of available information actuality and importance of this paper cannot be underestimated. The authors highlight the crucial issues such as interference of training workloads, intensity distribution, training periodization and fitness training. Emerging new concepts related to training, health status and injuries prevention were presented and considered. The proposed review has distinct value both for sport science and practice.
Thank you very much for your positive feedback and assessment of our triathlon manuscript.
Round 2
Reviewer 1 Report
All queries have been adressed.
Author Response
Thank you very much.